# Rethinking Softmax: Self-Attention with Polynomial Activations

## Abstract

This paper challenges the conventional belief that softmax attention in transformers is effective primarily because it generates a probability distribution for attention allocation. Instead, we theoretically show that its success lies in its ability to implicitly regularize the Frobenius norm of the attention matrix during training. We then explore alternative activations that regularize the Froebnius norm of the attention matrix, demonstrating that certain polynomial activations can achieve this effect, making them suitable for attention-based architectures. Empirical results indicate these activations perform comparably or better than softmax across various computer vision and language tasks, suggesting new possibilities for attention mechanisms beyond softmax.

## 1 Introduction

Transformer architectures (Vaswani et al., 2017) have become the state-of-the-art model architecture for diverse areas such as natural language processing (NLP) (Vaswani et al., 2017; Devlin et al., 2018; Zhuang et al., 2021; Zhen et al., 2022), computer vision (Dosovitskiy et al., 2020; Carion et al., 2020; Liu et al., 2021; Touvron et al., 2021), and robotics (Fu et al., 2024; Maiti et al., 2023; Salzmann et al., 2020). A key component in the transformer architecture is the softmax attention block, enabling transformers to evaluate the importance of individual input elements during output generation. This feature facilitates an efficient method to attend to diverse input elements throughout training, allowing transformers to effectively capture spatial dependencies within sequential data. Unlike traditional recurrent neural networks (RNNs) and convolutional neural networks (CNNs), transformers exhibit the ability to scale to large datasets without a significant degradation in performance. This characteristic has made them an ideal architecture for handling large-scale machine learning tasks.

Softmax is widely recognized for its effectiveness in attention mechanisms due to its ability to produce an attention matrix that meets three key conditions (Vaswani et al., 2017; Dosovitskiy et al., 2020; Zhen et al., 2022; AUEB et al., 2016): (i) non-negativity, (ii) rows that are normalized to sum to 1 and (iii) sparsity. The general consensus is that non-negativity guarantees that attention weights remain positive, facilitating the model in assigning significance to various input elements. The normalization constraint ensures that the attention weights for all input elements collectively sum to 1, rendering the weights interpretable as probabilities. Additionally, sparsity aids the model in focusing on a select few key elements, thereby enhancing efficiency. It has been argued that these properties are crucial for enabling the attention mechanism to attend to pertinent segments of the input sequence while efficiently filtering out irrelevant details. However, this approach to attention has become somewhat axiomatic as it is mostly motivated by empirical results with little theoretical foundation. Despite the exploration of alternative activations in several studies (Shen et al., 2023; Fang et al., 2022; Correia et al., 2019), softmax attention continues to dominate, largely due to its interpretability.

In this paper, we question this view by proposing that the effectiveness of softmax stems from its implicit regularization of the Frobenius norm of the attention matrix during training, preventing attention weights from becoming excessively large or small. We then derive a theoretical framework that produces polynomial activations that deliberately violate one or more of the three conditions mentioned earlier, yet are able to regularize the Frobenius norm of the attention weights during training. Our findings demonstrate that such activations can achieve comparable or even superior performance to softmax across various vision and natural language processing (NLP) tasks, even though they seem to violate our understanding of attention.

We advise the reader that this paper diverges from the usual pursuit of creating cutting-edge transformer architectures for achieving state-of-the-art results on benchmark datasets. Instead, our focus is on critically examining softmax attention to determine whether its effectiveness is a result of true interpretability or a more nuanced regularization mechanism. By questioning established views, we aim to uncover deeper insights into transformer architectures that could lead to broader applications and improved understanding. Nonetheless, we validate our theory on multiple transformer-based tasks, including image classification, segmentation, object detection, and NLP, often achieving results that match or surpass those of softmax attention. Our main contributions are:

1. We question the widely accepted notion that softmax's effectiveness in attention mechanisms is solely due to its ability to produce normalized sparse attention weights. Instead, we theoretically show that softmax has a regularization effect on attention and argue this plays a crucial role in its success.

2. We explore activations that deliberately deviate from traditional softmax attention conditions. These activations are found to regularize the Frobenius norm of the attention matrix during training, akin to softmax, and demonstrate comparable or superior performance across various vision and NLP tasks.

## 2 RELATED WORK

Several studies have explored alternative activations for attention mechanisms in transformers. Shen et al. (2023) investigated ReLU activations, finding them to outperform softmax in tasks with long sequences, such as document translation. Banerjee et al. (2020) examined Taylor series approximations to softmax, which showed superior performance to softmax in image classification. Wang et al. (2021) proposed periodic alternatives to softmax, designed to provide better gradients for attention mechanisms and achieved better results than softmax on simple networks for image classification. Koohpayegani & Pirsiavash (2024) demonstrated that applying $l^1$ normalization to linear attention mechanisms can achieve performance comparable to softmax. Our work differs from all these in that we identify a clear theoretical relationship between the scale of the Frobenius norm of the self-attention matrix and the input sequence length. Using this insight to derive potential acitvations that can perform on par with softmax.

## 3 PRELIMINARIES AND NOTATION

In this section we outline the definition of a transformer via the transformer block and set the notation of various mathematical quantities we will be using in future sections. For more details on transformers the reader can consult Vaswani et al. (2017); Dosovitskiy et al. (2020).

Transformer architectures comprise of transformer blocks, defined as follows. A transformer block is a mapping $\mathbf{T} : \mathbb{R}^{N \times D} \to \mathbb{R}^{N \times D}$ defined as

$$\mathbf{T}(x) = \mathbf{F}(\mathbf{A}(x) + x) \tag{3.1}$$

where $\mathbf{F}$ is a feedforward MLP with a residual connection and $\mathbf{A}$ is an attention head.

The attention head $\mathbf{A}$ is defined as follows: It comprises of three learnable matrices, a query ($q$), key ($k$) and value ($v$) defined by: $q = QX$, $k = KX$, $v = VX$ for an input sequence $X \in \mathbb{R}^{N \times D}$ with $Q, K \in \mathbb{R}^{D \times d}$ and $V \in \mathbb{R}^{D \times M}$. The attention head $\mathbf{A}(X)$ is then defined by

$$\mathbf{A}(X) = \phi(\mathcal{S}(q, k))v \tag{3.2}$$

where $\mathcal{S}$ is a similarity transformation and $\phi$ is an activation function. The most common used $\mathcal{S}$ is the dot-product: $\mathcal{S}(q, v) = qk^T$, known as self-attention, and will be the one we focus on in this paper. The most common activation function $\phi$ that is used by authors is softmax. This leads to the most common form of the attention head given by

$$\mathbf{A}(X) = \mathbf{softmax}\left(\frac{qk^T}{\sqrt{d}}\right)v = \mathbf{softmax}\left(\frac{XQK^TX^T}{\sqrt{d}}\right)XV. \tag{3.3}$$

The function **softmax** is the matrix softmax map that applies the usual softmax function row-wise:

$$\mathbf{softmax}\left(\begin{bmatrix} x_{11} & \cdots & x_{1n} \\ \vdots & \vdots & \vdots \\ x_{n1} & \cdots & x_{nn} \end{bmatrix}\right) = \begin{bmatrix} \frac{e^{x_{11}}}{\sum_{j=1}^n e^{x_{1j}}} & \cdots & \frac{e^{x_{1n}}}{\sum_{j=1}^n e^{x_{1j}}} \\ \vdots & \vdots & \vdots \\ \frac{e^{x_{n1}}}{\sum_{j=1}^n e^{x_{nj}}} & \cdots & \frac{e^{x_{nn}}}{\sum_{j=1}^n e^{x_{nj}}} \end{bmatrix} \tag{3.4}$$

The factor $\frac{1}{\sqrt{d}}$, as explained in Vaswani et al. (2017), is a scaling to prevent the gradients of softmax from being too small. For the theoretical analysis in this paper we will only use the dot-product similarity $qk^T$ and call the $N \times N$ matrix $softmax(qk^T)$ the *softmax self-attention matrix*. In the experiments section, Sec. 5, we will empirically validate our theoretical framework on more general softmax attention blocks.

For general transformer architectures, multiple heads $\mathbf{A}_i$ for $1 \leq i \leq n$ are used. Each attention head is defined by equation 3.3 and then all outputs of each attention head are concatenated together before going into the feedforward layer.

We will need notation for the derivative of the matrix softmax map defined by equation 3.4. Given a matrix $A \in \mathbb{R}^{N \times N}$ we can differentiate the matrix map **softmax** at $A$ and obtain the gradient linear map $\nabla\mathbf{softmax}(A) : \mathbb{R}^{N \times N} \to \mathbb{R}^{N \times N}$ that is defined by the formula

$$\nabla\mathbf{softmax}(A) := \mathbf{Jsoftmax}(A)^T \tag{3.5}$$

where $\mathbf{Jsoftmax}(A)$ is the Jacobian of **softmax** at $A$.

Given a matrix $A \in \mathbb{R}^{n \times m}$, we denote its Frobenius norm by $||A||_F$. Additionally, we use the notation $\mathbb{E}$ to represent the expectation of a random variable, where the specific random variable being considered will be clear from the context.

# 4 THEORETICAL ANALYSIS

## 4.1 IMPLICIT REGULATIZATION OF SOFTMAX

This section presents a theoretical result showing that the softmax activation imposes control over the Frobenius norm of the self-attention matrix in a way that grows sub-linearly with the input sequence's token length. Additionally, we demonstrate that the gradient of the softmax with respect to the self-attention matrix also exhibits a similar degree of regularity. While previous work has analyzed the regularity of softmax self-attention through the lens of the Lipschitz constant (Kim et al., 2021; Castin et al., 2023), our theorem offers a novel perspective by directly linking the Frobenius norm regularity to the token length. This provides insights into how self-attention activations should scale with token length to maintain stability during training, especially with gradient descent-based algorithms.

**Theorem 4.1.** *Let* $\mathbf{softmax} : \mathbb{R}^{N \times N} \to \mathbb{R}^{N \times N}$ *be the matrix softmax map defined by equation 3.4 and let* $\nabla\mathbf{softmax}(A) : \mathbb{R}^{N \times N} \to \mathbb{R}^{N \times N}$ *denote the gradient of* $\mathbf{softmax}$ *at* $A \in \mathbb{R}^{N \times N}$. *We then have the following bounds on the Frobenius norms*

$$||\mathbf{softmax}(A)||_F \leq \sqrt{N} \tag{4.1}$$

$$||\nabla\mathbf{softmax}(A)||_F \leq 2\sqrt{N}. \tag{4.2}$$

The key implication of theorem 4.1 is that during the training of a transformer with softmax self-attention, the Frobenius norm of each softmax self-attention matrix remains bounded by a value that grows as $\mathcal{O}(\sqrt{N})$. This ensures that backpropagation through the weights of the self-attention matrix does not lead to excessively large gradients. The proof hinges on the fact that the row normalization inherent in softmax effectively controls the Frobenius norm. For a detailed proof see appendix A.1.1.

## 4.2 POLYNOMIAL ACTIVATIONS FOR SELF-ATTENTION

In section 4.1, we demonstrated that softmax implicitly regularizes the Frobenius norm of the self-attention matrix. Building on this, we now show that by scaling specific polynomial activations, a similar regularization effect on the Frobenius norm can be achieved in expectation, closely replicating the impact of softmax.

**Theorem 4.2.** *Let $X \in \mathbb{R}^{N \times D}$ and $Q, K \in \mathbb{R}^{D \times d}$ be i.i.d random variables distributed according to $X \sim \mathcal{N}(0, \sigma_x)$ and $Q, K \sim \mathcal{N}(0, \sigma_t)$. We have the following expectations of the Frobenius norms of powers of the $N \times N$ matrix $(XQK^T X^T)^p$ for $p \geq 1$*

$$\mathbb{E}\left\|\left(\frac{XQK^T X^T}{\sqrt{d}}\right)^p\right\|_F \leq \mathcal{O}(N) \tag{4.3}$$

By scaling such an activation by $\frac{1}{\sqrt{N}}$ we can obtain a $\mathcal{O}(\sqrt{N})$ bound.

**Corollary 4.3.** *Assume the same conditions as in theorem 4.2. Then*

$$\mathbb{E}\left\|\frac{1}{\sqrt{N}}\left(\frac{XQK^T X^T}{\sqrt{d}}\right)^p\right\|_F \leq \mathcal{O}(\sqrt{N}). \tag{4.4}$$

Corollary 4.3 establishes that activations of the form $\phi(x) := \frac{1}{\sqrt{N}} x^p$ provide a level of regularization, in expectation, similar to that of softmax when applied to the self-attention matrix. The proof of theorem 4.2 can be found in appendix A.1.2. The next property we want to prove is one similar to the gradient bound obtained in theorem 4.1. Since the self-attention matrix has parameters given by the queries $Q$ and keys $K$ (Vaswani et al., 2017), this implies that during the training of a transformer the $Q$ and $K$ matrices are the only aspects of the self-attention matrix that get updated. Therefore, we compute a regularity result with respect to the $Q$ and $K$ derivatives.

**Theorem 4.4.** *Let $X \in \mathbb{R}^{N \times D}$ and $Q, K \in \mathbb{R}^{D \times d}$ be i.i.d random variables distributed according to $X \sim \mathcal{N}(0, \sigma_x)$ and $Q, K \sim \mathcal{N}(0, \sigma_t)$. Then the expectation of the of the derivative of the matrix $\frac{(XQK^T X^T)^p}{\sqrt{d}}$ w.r.t the $Q$ parameter matrix for $p \geq 1$ is given by*

$$\mathbb{E}\left\|\frac{\partial}{\partial Q}\left(\frac{(XQK^T X^T)^p}{\sqrt{d}}\right)\right\| \leq \mathcal{O}(N) \tag{4.5}$$

The above theorem then suggests that if we scale the polynomial $x \to x^p$ by $\frac{1}{\sqrt{N}}$ the $Q$ derivative will have $\mathcal{O}(\sqrt{N})$ growth.

**Corollary 4.5.** *Assume the same condition as in theorem 4.4. Then*

$$\mathbb{E}\left\|\frac{1}{\sqrt{N}}\frac{\partial}{\partial Q}\left(\frac{(XQK^T X^T)^p}{\sqrt{d}}\right)\right\| \leq \mathcal{O}(\sqrt{N}). \tag{4.6}$$

An analogous estimate holds for derivatives with respect to the $K$ matrix. The proof of theorem 4.4 can be found in appendix A.1.2.

Corollaries 4.3 and 4.5 suggest that polynomial activations of the form $\phi(x) = \frac{1}{\sqrt{N}} x^p$, with $p > 0$, can achieve performance comparable to softmax when applied to self-attention matrices. In section 5, we empirically compare these activations to softmax and observe that they outperform softmax on a variety of transformer tasks. We focus on $p = 1$ and $p = 3$ as these polynomials clearly violate key aspects of softmax based attention, such as normalized rows, positivity, and sparsity. For larger values of $p$, performance declines due to the functions $\phi(x) = \frac{1}{\sqrt{N}} x^p$ having smaller gradients around 0 when $p$ is large, causing difficulties in training.

## 5 EXPERIMENTS

In this section, we validate the theory from section 4 on a variety of transformer tasks. We perform the empirical validation on two primary activations from section 4, namely a cubic polynomial activation $\phi(x) = x^3$ and a linear polynomial $\phi(x) = x$. The goal will be to show that by suitably scaling these activations using the theory in section 4, we can achieve competitive performance when compared to softmax. For the rest of this section we will simply denote these activations by $x^3$ and $x$.

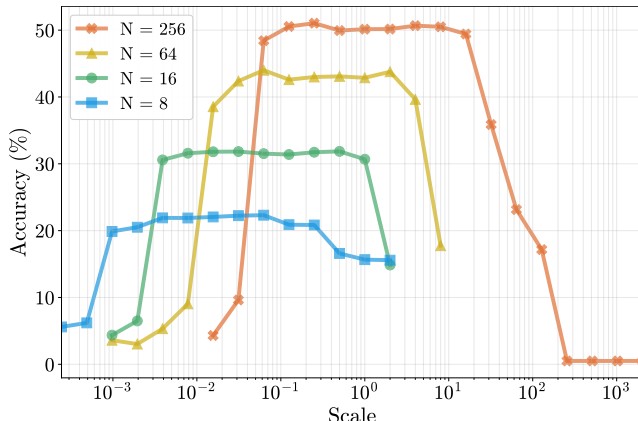

Figure 1: Training ViT-Tiny with the activation $\phi(x) = x^3$ with different sequence lengths and different scales. As the sequence length gets smaller, the log scale needed to obtain good accuracy decreases validating the theory from section 4.2.

## 5.1 IMAGE CLASSIFICATION

### 5.1.1 VIT-TINY ON TINY-IMAGENET:

In this section we test the theory from section 4 on the ViT-Tiny architecture (Steiner et al., 2021) trained from scratch on the Tiny-Imagenet dataset (Le & Yang, 2015).

Our first experiment was to test how the Top-1% accuracy changes for a ViT-Tiny trained on Tiny-Imagenet as we change the sequence length of the input and the scale predicted in corollaries 4.3 and 4.5 when using the activation $x^3$. According to the theory developed in section 4, the Frobenius norm scales according to $\mathcal{O}(\sqrt{N})$ when we scale $X^3$ by $\frac{1}{\sqrt{N}}$. Thus as the sequence length decreases we should see the amount of scaling in a Log scale decrease.

Figure 1 shows the results of this experiment. We considered four different input sequence lengths of sizes 256, 64, 16 and 8. We ran several ViT-Tiny architectures with a variety of scalings of the form $\mathcal{O}(\frac{1}{\sqrt{N}})$ where $N$ ranged below to above the sequence length. As can be seen from figure 1 as the sequence length got smaller the amount of scaling, shown in Log scale on the x-axis, needed for good accuracy got smaller verifying the theory in section 4.2.

The second experiment compared activations $x^3$ and $x$, along with scaled versions $\frac{1}{16}x^3$ and $\frac{1}{16}x$, against softmax using the Tiny-ViT architecture on Tiny-Imagenet. With a sequence length of 256 ($\sqrt{256} = 16$), we decided to take $\frac{1}{16}$ as the scale of the polynomial activations. The experiment used a patch size of 4, 3 attention heads, and 12 layers as described in Steiner et al. (2021). Results in table 1 show $\frac{1}{8}x^3$ outperforming softmax, while the unscaled version performed poorly. Similarly, $\frac{1}{16}x$ performed competitively with a significant drop in performance without scaling.

Figure 2 displays the Frobenius norm of the self-attention matrix during training for five activations in layers 2 and 12 of ViT-Tiny, averaged across all heads. Norms for $x^3$ and $x$ are higher than softmax, but scaling by $\frac{1}{16}$ reduces them to more stable levels, improving training stability. Similarly, figure 3 shows the Jacobian's Frobenius norm, where scaling also brings the norms closer to softmax, ensuring more stable gradients. Further plots for other layers are in appendix A.2.2.

|  | softmax | $\frac{x^3}{16}$ | $x^3$ | $\frac{x}{16}$ | $x$ |
|---|---|---|---|---|---|
| Top-1% accuracy | 50.26 | **50.5** | 45.3 | 47.9 | 31.78 |

Table 1: Comparison of Top-1% accuracy on Tiny-Imagenet between softmax and polynomial activations. The cubic activation outperforms softmax when the right scale of $\frac{1}{8}$ is applied. Similarly, the linear activation is competitive only with an optimal scale.

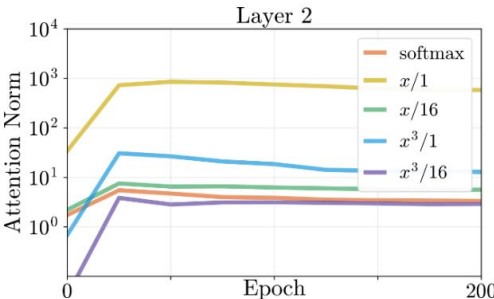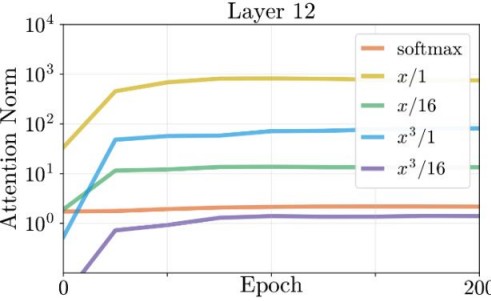

Figure 2: Frobenius norm of the self-attention matrix with five different activations in layer 2 and 12 of the ViT-Tiny architecture during training.

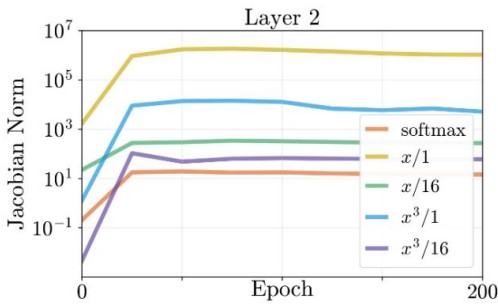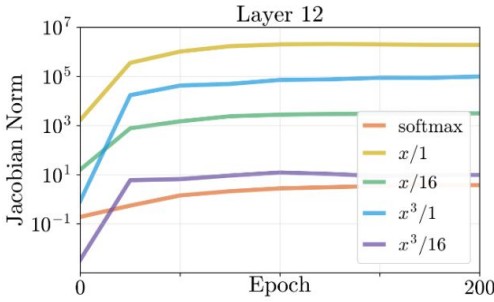

Figure 3: Frobenius norm of the Jacobian of the self-attention matrix with five different activations in layer 2 and 12 of the ViT-Tiny architecture during training.

### 5.1.2 LARGER VISION TRANSFORMERS ON IMAGENET-1K

For this experiment we carried out an image classification task using a variety of different vision transformers from the literature on the ImageNet-1k dataset. We found that the scales $\frac{1}{14}$ worked best for both $x^3$ and $x$.

We train all models on the ImageNet-1k dataset from scratch and report Top-1 accuracy on the validation set. We use PyTorch Paszke et al. (2019) and Timm Wightman (2019) libraries to train our models with similar setups to He et al. (2022) Liu et al. (2021). We examined our approach along with the following three transformer architectures to show its generalization:

- **ViT**: Dosovitskiy et al. (2020) is the pioneering work that interprets an image as a sequence of patches and processes it by a standard Transformer encoder as used in NLP. This simple, yet scalable, strategy works surprisingly well when coupled with pre-training on large datasets. We use ViT-Small which has the following settings: patch size = 16, embedding dimensions = 384, number of heads = 6, and layers = 12. Also, We use ViT-Base which has the following settings: patch size = 16, embedding dimensions = 768, number of heads = 12, and layers = 12.

- **DeiT**: Touvron et al. (2021) is a well-known transformer based on ViT. It is very similar to ViT except that it converged faster due to better training recipe since we do not use the distillation token imposed in DeiT. We use DeiT-Small which has the following settings: patch size = 16, embedding dimensions = 384, number of heads = 6, and layers = 12. Also, We use Deit-Base which has the following settings: patch size = 16, embedding dimensions = 768, number of heads = 12, and layers = 12.

- **Swin Transformer**: Liu et al. (2021) produces a hierarchical feature representation and proposes the shifted window based self-attention which is shown to be effective and efficient on vision problems. We use Swin-Small with 96 channels and Swin-Base with 128 channels in the hidden layers of the

first stage. The window size is set to $M = 7$ by default, the query dimension of each head is $d = 32$, and layer numbers are $2, 2, 18, 2$ for all experiments.

**- XciT**: Xiong et al. (2021) is a vision transformer architecture consisting of two different components compared to the standard ViT. Firstly, it has Local Patch Interaction in each block, which includes one depth-wise 3×3 convolution followed by Batch Normalization, GELU, and another depth-wise 3×3 convolution. Secondly, it uses Cross-Covariance attention, where the attention map is derived from the cross-covariance matrix computed over the key and query projections of the token features. We use XCiT-S12 with a patch size of 16 and XCiT-M24 with a patch size of 24.

The results are shown in table 2. The activation $\frac{1}{14}x^3$ performed best on the ViT's and the Swin transformers while softmax performed best on the DeiT architectures. Further ablations with different scales and activations can be fond in appendix A.2.1.

| Activation | Models | | | | | | | |
| --- | --- | --- | --- | --- | --- | --- | --- | --- |
| | ViT-Base | ViT-Small | DeiT-Base | DeiT-Small | Swin-Base | Swin-Small | XciT-Medium | XciT-Small |
| softmax | **79.6** | 80.2 | 78.9 | 79.6 | 83.0 | 83.3 | 81.1 | 81.2 |
| $\frac{x^3}{14}$ | **79.6** | **80.5** | 77.4 | 78.3 | **83.2** | **83.4** | **81.2** | **82.1** |
| $x^3$ | 77.8 | 78.6 | 76.1 | 76.3 | 79.7 | 79.9 | 78.1 | 78.3 |
| $\frac{x}{14}$ | 76.9 | 77.8 | **79.6** | **79.8** | 79.4 | 79.5 | 79.2 | 79.3 |
| $x$ | 73.2 | 73.9 | 77.7 | 77.9 | 77.8 | 77.9 | 76.4 | 76.6 |

Table 2: Comparsions of pre-training models with different activation functions on ImageNet-1k datasets. We report the classification top-1 accuracy (%).

Figure 4 plots the Frobenius norm of the self-attention matrix with the five different activations during training in layers 2 and 12, averaged over all heads within each layer, of the ViT-Small architecture. By scaling the activations $x^3$ and $x$ by $\frac{1}{14}$ we were able to control the scale of the Frobenius norm of the self-attention matrix and obtain scales comparable to softmax's scale. Simiarly, figure 5 plots the Frobenius norm of the Jacobian of the self-attention matrix during training for layers 2 and 12, averaged over all heads. By scaling the activations $x^3$ and $x$ by $\frac{1}{14}$ we were able to control the scale of the Jacobian norm and obtain scales comparable to softmax's scale during training. Plots for other layers of the architecture during training are given in appendix A.2.2.

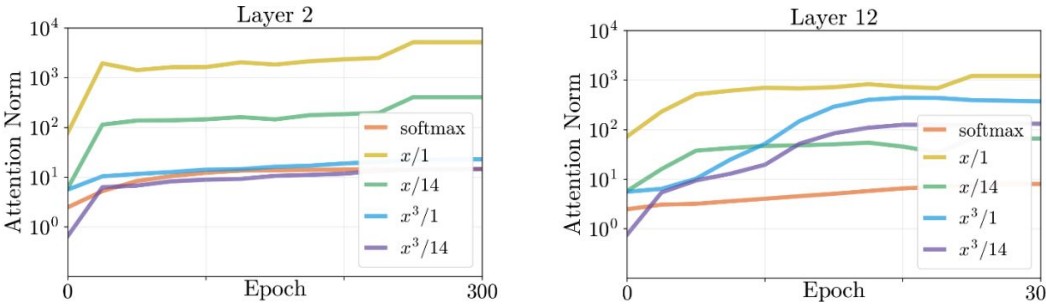

Figure 4: Frobenius norm of the self-attention matrix with five different activations in layer 2 and 12 of the ViT-Tiny architecture during training.

### 5.1.3 VISUALIZING SELF-ATTENTION WITH VIT-BASE

We plotted heat maps of self-attention matrices using the $\frac{1}{14}x^3$ and softmax activations across two layers and heads after convergence, averaging over a fixed training batch of size 128. Figure 6 shows layer 2, head 8, highlighting the differences in attention patterns, with $\frac{1}{14}x^3$ containing both positive and negative values. Similarly, Figure 7 for layer 12, head 6 shows distinct patterns for each activation. Overall, the ViT-Base architecture with $\frac{1}{14}x^3$ exhibits notably different self-attention patterns compared to softmax.

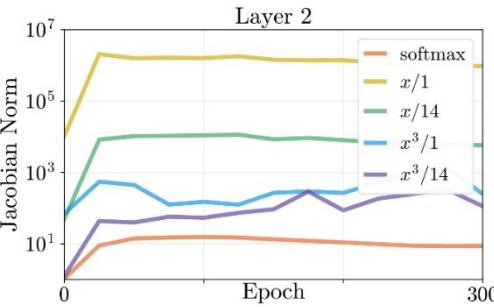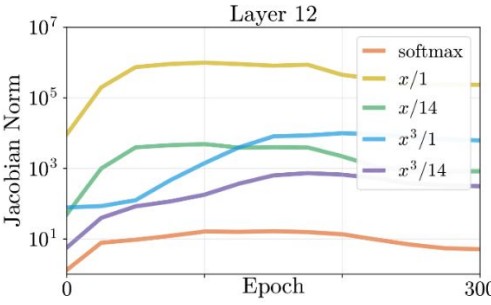

Figure 5: Frobenius norm of the Jacobian of the self-attention matrix with five different activations in layer 2 and 12 of the ViT-Tiny architecture during training.

We visualized how the self-attention matrix targets different regions of an image. Using an image from the ImageNet-1k validation set, we extracted the class token, reshaped it into a $(14, 14)$ grid representing 196 patches, and then mapped it back to the original image size with nearest neighbor interpolation. Figure 8 shows the input image, while figure 9 illustrates the self-attention matrices for different activations in layer 12, head 6 of the ViT-Base architecture after convergence, highlighting their distinct focus areas.

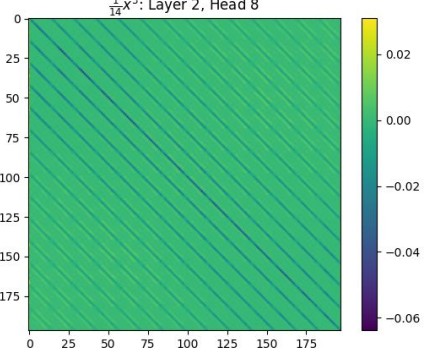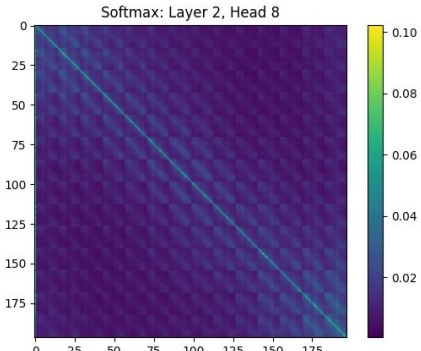

Figure 6: Heat maps of the self-attention matrix in layer 2, head 8, of a ViT base architecture, comparing $\frac{1}{14}x^3$ (left) and softmax (right) activations after training. The stark difference in self-attention patterns between the two activations is evident, showing distinct distributions across input tokens.

### 5.2 OBJECT DETECTION AND INSTANCE SEGMENTATION

In this section, in order to examine the transfer learning ability of our models, we demonstrate our approach to object detection and segmentation tasks by fine-tuning our ImageNet-pretrained XCiT model on them. Our experiments are conducted on COCO 2017Lin et al. (2014), which has 118K training images and 5K validation images with 80 categories. We integrate the XCiT architecture as the backbone in the Mask R-CNN (He et al., 2017) detector with a Feature Pyramid Network (FPN). Due to XCiT's inherently columnar design, we adapt it for FPN compatibility by extracting features from various layers for XCiT-S12. These features have a consistent stride of either 16. The feature resolutions are then adjusted to strides of [4, 8, 16, 32] This downsampling is accomplished through max pooling, while upsampling is achieved using a single transposed convolution layer. The model is trained for 36 epochs using the AdamW optimizer with learning rate of $10^{-4}$, 0.05 weight decay and 16 batch size. In table 3,we condunct experiments on XCiT-S12 models using 16×16 patches with the activations $\frac{1}{14}x^3$, $\frac{1}{14}x$ and softmax. We found we couldn't train with the activations $x^3$ and $x$ on this task well so only report the others.

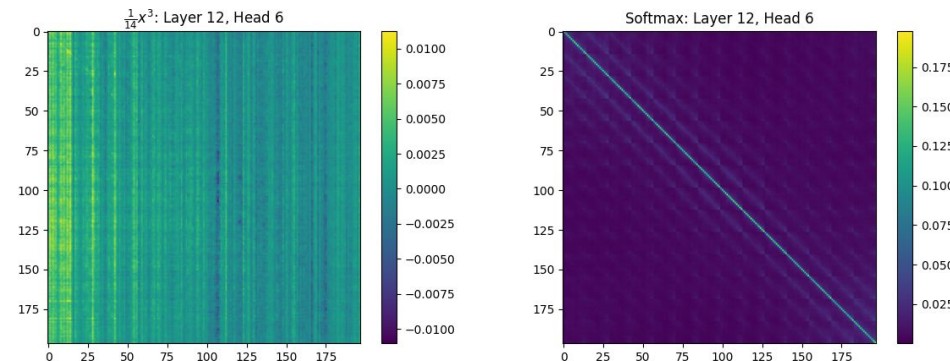

Figure 7: Heat maps of the self-attention matrix in layer 12, head 6, of a ViT base architecture, comparing $\frac{1}{14}x^3$ (left) and softmax (right) activations after training. The contrast in self-attention patterns between the two activations is clearly visible.

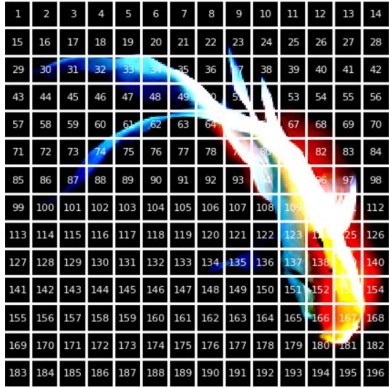

Figure 8: Fish image from validation set of ImageNet-1k broken up into $196 = 14 \times 14$ patches.

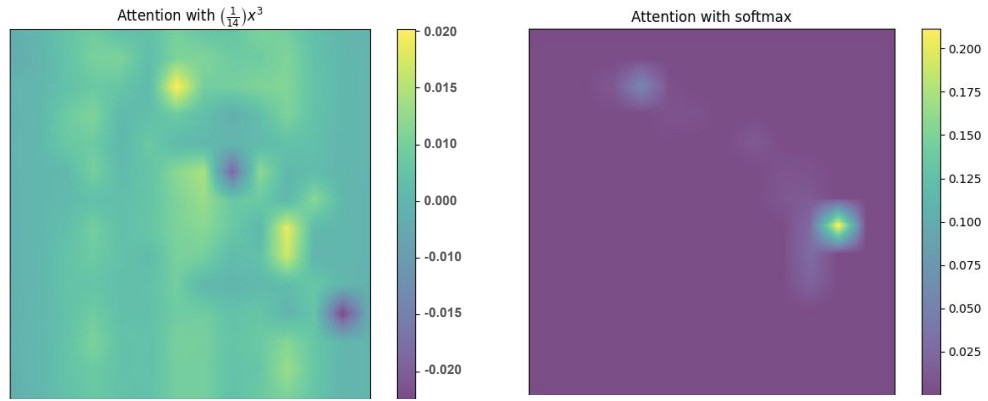

Figure 9: Comparing how $\frac{1}{14}x^3$ and softmax self-attention matrices focus on different parts of the fish image from figure 8 in layer 12, head 6, after the model has converged.

| Activation | $AP^b$ | $AP^b_{50}$ | $AP^b_{75}$ | $AP^m$ | $AP^m_{50}$ | $AP^m_{75}$ |
|---|---|---|---|---|---|---|
| softmax | **44.9** | 66.1 | 48.9 | **40.1** | **63.1** | 42.8 |
| $\frac{x^3}{14}$ | 44.8 | **66.3** | **49** | **40.1** | **63.1** | **42.9** |
| $\frac{x}{14}$ | 44.8 | 66.2 | **49** | **40.1** | **63.1** | 42.8 |

Table 3: COCO object detection and instance segmentation performance on the mini-val set. All backbones are pretrained on ImageNet-1k, and use Mask R-CNN model. $AP^b$: Average Precision for bounding box predictions, $AP^b_{50/75}$: Average Precision at an Intersection over Union (IoU) threshold of 0.50/0.75 for bounding box predictions, $AP^m$: Average Precision for mask predictions, $AP^m_{50/75}$: Average Precision at an Intersection over Union (IoU) threshold of 0.50/0.75 for mask predictions

## 5.3 NATURAL LANGUAGE PROCESSING(NLP)

To assess the effectiveness of our approach on NLP tasks, we trained models on five benchmarks from the Long Range Arena (LRA) suite Tay et al. (2020): **ListOps**, **Text Classification**, **Retrieval**, **Image Classification**, and **Pathfinder**. We evaluated the activations $\frac{x^3}{14}$ and $\frac{x}{14}$ against softmax, finding that $x^3$ and $x$ did not train effectively on their own, so only results for these scaled activations and softmax are presented. Our implementation followed the guidelines from Xiong et al. (2021). The results are summarized in table 4.

| Activation | ListOps | Text | Retrieval | Image | Pathfinder |
|---|---|---|---|---|---|
| softmax | 37.1 | **63.8** | 79.8 | **39.9** | **72.9** |
| $\frac{x^3}{14}$ | **37.5** | 63.3 | 80.9 | 37.2 | 68.7 |
| $\frac{x}{14}$ | 34.3 | 62.9 | **81.5** | 39.0 | 69.1 |

Table 4: Comparsions of transformer models with different activation functions on NLP tasks. We report the accuracy (%) on LRA benchmarks.

## 6 LIMITATIONS

While our work introduces novel activations that challenge the conventional softmax approach, there are some limitations to address. Our theoretical framework is primarily designed for dot-product self-attention and may not immediately extend to other attention mechanisms, although our empirical results showed competitive performance against softmax across different architectures. Additionally, we observed that while our activations performed well on vision tasks, their performance was less consistent on NLP tasks, suggesting that a more refined theoretical approach may be needed for these applications.

## 7 CONCLUSION

This work challenges the traditional view that transformer activations for attention must produce sparse probability distributions. We introduced a theoretical framework analyzing the Frobenius norm of the self-attention matrix, which suggests key scaling properties for activations in attention mechanisms. We proved that specific polynomial activations, which behave very differently from softmax, satisfy these properties. Through extensive experiments across vision and NLP tasks, we demonstrated that these alternative activations not only compete with but sometimes outperform softmax, offering a fresh perspective on attention mechanisms in transformers.

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

# A APPENDIX

## A.1 THEORETICAL ANALYSIS

### A.1.1 PROOFS FOR THEOREMS IN SECTION 4.1

In this section we give the proof of theorem 4.1.

*Proof of theorem 4.1.* We will start by proving the first inequality in theorem 4.1. Given a matrix $A = (a_{ij}) \in \mathbb{R}^{N \times N}$ we have that

$$
\mathbf{softmax}\left(\begin{bmatrix} a_{11} & \cdots & a_{1n} \\ \vdots & \vdots & \vdots \\ a_{n1} & \cdots & a_{nn} \end{bmatrix}\right) = \begin{bmatrix} \frac{e^{a_{11}}}{\sum_{j=1}^n e^{a_{1j}}} & \cdots & \frac{e^{a_{1n}}}{\sum_{j=1}^n e^{a_{1j}}} \\ \vdots & \vdots & \vdots \\ \frac{e^{a_{n1}}}{\sum_{j=1}^n e^{a_{nj}}} & \cdots & \frac{e^{a_{nn}}}{\sum_{j=1}^n e^{a_{nj}}}. \end{bmatrix} \tag{A.1}
$$

By definition of the Frobenius norm we then see that

$$
||\mathbf{softmax}(A)||_F^2 = \left(\frac{1}{\sum_{j=1}^n e^{a_{1j}}}\right)^2 (e^{2a_{11}} + \cdots e^{2a_{11}}) + \cdots + \left(\frac{1}{\sum_{j=1}^n e^{a_{Nj}}}\right)^2 (e^{2a_{N1}} + \cdots e^{2a_{NN}}) \tag{A.2}
$$

$$
\leq \left[\left(\frac{1}{\sum_{j=1}^n e^{a_{1j}}}\right)(e^{a_{11}} + \cdots e^{a_{11}})\right]^2 + \cdots + \left[\left(\frac{1}{\sum_{j=1}^n e^{a_{Nj}}}\right)(e^{a_{N1}} + \cdots e^{a_{NN}})\right]^2 \tag{A.3}
$$

$$
= 1 + \cdots + 1 \tag{A.4}
$$

$$
= N \tag{A.5}
$$

where the second inequality uses the fact that for non-negative numbers $a$ and $b$ we always have that $a^2 + b^2 \leq (a + b)^2$.

It then immediately follows that $||\mathbf{softmax}(A)||_F \leq \sqrt{N}$ and this proves the first inequality in the statement of theorem 4.1.

We move on to prove the second inequality in the statement of theorem 4.1. For this, let us write each entry of the matrix on the right of equation A.1 as follows:

$$
F_{kl} = \frac{e^{a_{kl}}}{\sum_{j=1}^N e^{a_{kj}}}. \tag{A.6}
$$

By applying the chain rule we then have the following derivative formulas

$$
\frac{\partial}{\partial x_{ij}} F_{ij} = F_{ij} - F_{ij}^2 \tag{A.7}
$$

$$
\frac{\partial}{\partial x_{ik}} F_{ij} = -F_{ij} F_{ik} \text{ for any } k \neq j \tag{A.8}
$$

$$
\frac{\partial}{\partial x_{kl}} F_{ij} = 0 \text{ for any } k \neq i \text{ and } l \neq j. \tag{A.9}
$$

We can then express the gradient as

$$
\nabla\mathbf{softmax}(A) = \begin{bmatrix} \nabla F_{11} \\ \vdots \\ \nabla F_{1N} \\ \nabla F_{21} \\ \vdots \\ \vdots \\ \nabla F_{NN} \end{bmatrix} \tag{A.10}
$$

where
$$\nabla F_{ij} = \begin{bmatrix} -F_{ij}F_{i1}^2 & -F_{ij}F_{i2} & \cdots & F_{ij} - F_{ij}^2 & \cdots & F_{ij}F_{iN}. \end{bmatrix} \tag{A.11}$$
From these computations we see that
$$||\nabla \mathbf{softmax}(A)||_F^2 = ||\nabla F_{11}||_F^2 + \cdots + ||\nabla F_{NN}||_F^2. \tag{A.12}$$
We will proceed by bounding each collection $||\nabla F_{i1}||_F^2 + \cdots + ||\nabla F_{1N}||_F^2$ separately then add up all the bounds. We have

$$||\nabla F_{i1}||_F^2 + \cdots + ||\nabla F_{1N}||_F^2 = |F_{i1} - F_{i1}^2|^2 + |F_{i1}F_{i2}|^2 + \cdots + |F_{i1}F_{iN}|^2 \tag{A.13}$$
$$+ |F_{i2}F_{i1}|^2 + |F_{i2} - F_{i2}^2|^2 + \cdots + |F_{i2}F_{iN}|^2 \tag{A.14}$$
$$+ \cdots\cdots\cdots + \tag{A.15}$$
$$+ |F_{iN}F_{i1}|^2 + |F_{iN}F_{i2}|^2 + \cdots + |F_{iN} - F_{iN}^2|^2 \tag{A.16}$$
$$\leq (F_{i1})^2(|1 - F_{i1}| + |F_{i2}| + \cdots + |F_{iN}|)^2 \tag{A.17}$$
$$(F_{i2})^2(|F_{i1}| + |1 - F_{i2}| + \cdots + |F_{iN}|)^2 \tag{A.18}$$
$$+ \cdots\cdots\cdots + \tag{A.19}$$
$$+ (F_{iN})^2(|F_{i1}| + |F_{i2}| + \cdots + |1 - F_{iN}|)^2. \tag{A.20}$$

We then observe that since $F_{i1} + \cdots + F_{iN} = 1$ we have that $1 - F_{ij} = 2(F_{i1} + \cdots + \widehat{F_{ij}} + \cdots + F_{iN})$ where $\widehat{F_{ij}}$ means we don't include $F_{ij}$ in the sum. This means we get the bound

$$||\nabla F_{i1}||_F^2 + \cdots + ||\nabla F_{1N}||_F^2 \leq 4F_{i1}^2(\widehat{F_{i1}} + F_{i2} + \cdots + F_{iN}) \tag{A.21}$$
$$+ \cdots\cdots\cdots + \tag{A.22}$$
$$+ 4F_{iN}^2(F_{i1} + F_{i2} + \cdots + \widehat{F_{iN}}) \tag{A.23}$$
$$\leq 4(F_{i1}^2 + \cdots F_{iN}^2) \tag{A.24}$$
$$= 4. \tag{A.25}$$

Putting all the bounds together for each of the terms $N$ terms $||\nabla F_{i1}||_F^2 + \cdots + ||\nabla F_{1N}||_F^2$ we get
$$||\nabla \mathbf{softmax}(A)||_F^2 \leq 4N \tag{A.26}$$
and this implies
$$||\nabla \mathbf{softmax}(A)||_F \leq 2\sqrt{N}. \tag{A.27}$$
This finishes the proof of theorem 4.1.

$\square$

### A.1.2 PROOFS FOR THEOREMS SECTION 4.2

In this section we will give the proof of theorems 4.2 and 4.4.

*Proof of theorem 4.2.* We will split the matrix product $XQK^TX^T$ and think of it as the product of two matrices. Suppose $\mathbf{A} \in \mathbb{R}^{N \times D} \sim \mathcal{N}(0, \sigma_1^2)$, $\mathbf{B} \in \mathbb{R}^{D \times N} \sim \mathcal{N}(0, \sigma_2^2)$ and $\mathbf{C} = \mathbf{AB}$. Each element in the matrix $\mathbf{C}$ can be written as a product of a row of $\mathbf{A}$ with a column of $\mathbf{B}$. Since expectation is linear, we need to compute the expectation of each of these elements. We do the case of the entry $c_{11}$ which is the entry in $\mathbf{C}$ in the first row and first column. For the $p = 1$ case we can then compute

$$\mathbb{E}(c_{11}^2) = \mathbb{E}((\sum_{i=1}^{D} a_{1i}b_{i1})^2)$$
$$= \mathbb{E}(\sum_{i=1}^{D} a_{1i}^2 b_{i1}^2 + \sum_{i=1}^{D}\sum_{j=1,j\neq i}^{D} a_{1i}b_{i1}a_{1j}b_{j1}) \tag{A.28}$$
$$= \sum_{i=1}^{D} \mathbb{E}(a_{1i}^2)\mathbb{E}(b_{i1}^2) + \sum_{i=1}^{D}\sum_{j=1,j\neq i}^{D} \mathbb{E}(a_{1i})\mathbb{E}(b_{i1})\mathbb{E}(a_{1j})\mathbb{E}(b_{j1})$$
$$= D\sigma_1^2\sigma_2^2 + 0.$$

The Frobenius norm of the matrix $\mathbf{C}$ is just the sum of these values for all $N^2$ elements and this proves the $p = 1$ case.

For the case that $p > 1$ we proceed in a similar way. The key observation is that odd powers, in the matrix expansion, will have expectaion $0$, so we need only consider the even powers. Therefore, suppose $\mathbf{C} = (\mathbf{AB})^p$. We will compute the expectation of the first entry $c_11 \in \mathbf{C}$:

$$
\begin{aligned}
\mathbb{E}(c_{11}^2) &= \mathbb{E}((\sum_{i=1}^{D} a_{1i} b_{i1})^{2p}) \\
&= \mathbb{E}(\sum_{i=1}^{D} a_{1i}^{2p} b_{i1}^{2p} + \sum_{i=1}^{D} \sum_{j=1, j\neq i}^{D} a_{1i}^{2p-2} b_{i1}^{2p-2} a_{1j}^2 b_{j1}^2 + \cdots).
\end{aligned}
\tag{A.29}
$$

Note that the first term only has a count of $D$ and the second term has a count of $D(D-1)$. Thus, we only need to consider the $\mathcal{O}(D^p)$ term where all the components have a power of $2$. The count is similar to choosing $p$ items from $D$,

$$
\begin{aligned}
\mathbb{E}(c_{11}^2) &\approx \mathbb{E}(\sum_{\{i_1,\ldots,i_p\}\in\{1,\ldots,D\}} \prod_{k=1}^{p} a_{1,i_k}^2 b_{i_k,1}^2) \\
&= \binom{D}{p} \frac{2p!}{2^p} \sigma_1^{2p} \sigma_2^{2p} \\
&= \frac{D!}{(D-p)!} \frac{2p!}{p!2^p} \sigma_1^{2p} \sigma_2^{2p} \\
&= \frac{D!}{(D-p)!} \frac{2p!}{2p!!} \sigma_1^{2p} \sigma_2^{2p} \\
&= \frac{D!}{(D-p)!} (2p-1)!! \sigma_1^{2p} \sigma_2^{2p}.
\end{aligned}
\tag{A.30}
$$

$\frac{D!}{(D-p)!}$ can always be bounded above by $D^p$, so the expectation can be upper bounded by $D^p(2p-1)!!\sigma_1^{2p}\sigma_2^{2p}$ and thus we get a quantity of the form $\mathcal{O}(N)$.

$\square$

*Proof of theorem 4.4.* We will do the $p = 1$ case first. We proceed similar to the proof of Theorem 4.2.

$$\mathbb{E}(\|\frac{\partial XQK^T X^T}{\partial Q}\|_F^2) = \sum_{i=1}^{N}\sum_{j=1}^{N}\mathbb{E}(|\frac{\partial x_i^T QK^T x_j}{\partial Q}|_F^2) \tag{A.31}$$

$$= \sum_{i=1}^{N}\sum_{j=1}^{N}\mathbb{E}(\|x_i x_j^T K\|_F^2) \tag{A.32}$$

$$= \sum_{i=1}^{N}\sum_{j=1}^{N}\mathbb{E}(\sum_{k=1}^{D}\sum_{l=1}^{d}(x_{ik}\sum_{m=1}^{D}x_{jm}k_{ml})^2) \tag{A.33}$$

$$= \sum_{i=1}^{N}\sum_{j=1}^{N}\mathbb{E}(\sum_{k=1}^{D}\sum_{l=1}^{d}x_{ik}^2(\sum_{m=1}^{D}x_{jm}k_{ml})^2) \tag{A.34}$$

$$= \sum_{i=1}^{N}\sum_{j=1}^{N}\mathbb{E}(\sum_{k=1}^{D}\sum_{l=1}^{d}x_{ik}^2(\sum_{m=1}^{D}x_{jm}^2 k_{ml}^2 + \sum_{m=1}^{D}\sum_{n=1,n\neq m}^{D}x_{jm}k_{ml}x_{jn}k_{nl})) \tag{A.35}$$

$$= \sum_{i=1}^{N}\sum_{j=1}^{N}\sum_{k=1}^{D}\sum_{l=1}^{d}(\sum_{m=1}^{D}\mathbb{E}(x_{ik}^2 x_{jm}^2 k_{ml}^2) + \sum_{m=1}^{D}\sum_{n=1,n\neq m}^{D}\mathbb{E}(x_{ik}^2 x_{jm}k_{ml}x_{jn}k_{nl})) \tag{A.36}$$

$$= \sum_{i=1}^{N}\sum_{j=1}^{N}\sum_{k=1}^{D}\sum_{l=1}^{d}(\sum_{m=1}^{D}\mathbb{E}(x_{ik}^2 x_{jm}^2 k_{ml}^2) + 0) \tag{A.37}$$

$$= \sum_{i=1}^{N}\sum_{j=1}^{N}\sum_{k=1}^{D}\sum_{l=1}^{d}\sum_{m=1}^{D}\mathbb{E}(x_{ik}^2 x_{jm}^2 k_{ml}^2) \tag{A.38}$$

$$= \sum_{i=1}^{N}\sum_{k=1}^{D}\sum_{l=1}^{d}\mathbb{E}(x_{ik}^2 x_{ik}^2 k_{kl}^2) + \sum_{i=1}^{N}\sum_{j=1,j\neq i}^{N}\sum_{k=1}^{D}\sum_{l=1}^{d}\sum_{m=1,m\neq k}^{D}\mathbb{E}(x_{ik}^2 x_{jm}^2 k_{ml}^2) \tag{A.39}$$

$$= NDd3\sigma_x^4\sigma_w^2 + N(N-1)D(D-1)d\sigma_x^4\sigma_w^2 \tag{A.40}$$

$$\approx N^2 D^2 d\sigma_x^4\sigma_w^2. \tag{A.41}$$

When $p > 1$ we can proceed in a similar way.

$$\mathbb{E}(\|\frac{\partial(XQK^T X^T)^p}{\partial Q}\|_F^2) = \sum_{i=1}^{N}\sum_{j=1}^{N}\mathbb{E}(\|\frac{(\partial x_i^T QK^T x_j)^p}{\partial Q}\|_F^2) \tag{A.42}$$

$$= \sum_{i=1}^{N}\sum_{j=1}^{N}\mathbb{E}(\|p(x_i^T QK^T x_j)^{p-1}\frac{\partial x_i^T QK^T x_j}{\partial Q}\|_F^2) \tag{A.43}$$

$$= \sum_{i=1}^{N}\sum_{j=1}^{N}\mathbb{E}(\|p(x_i^T QK^T x_j)^{p-1}x_i x_j^T K\|_F^2) \tag{A.44}$$

$$= \sum_{i=1}^{N}\sum_{j=1}^{N}\mathbb{E}(p^2(x_i^T QK^T x_j)^{2p-2}\sum_{k=1}^{D}\sum_{l=1}^{d}(x_{ik}\sum_{m=1}^{D}x_{jm}k_{ml})^2). \tag{A.45}$$

We know that

$$(x_i^T Q K^T x_j)^{2p-2} = (\sum_{l=1}^{d}((\sum_{k=1}^{D} x_{ik} q_{kl}) \cdot (\sum_{m=1}^{D} x_{jm} k_{ml})))^{2p-2} \tag{A.46}$$

$$= (\sum_{l=1}^{d} \sum_{k=1}^{D} \sum_{m=1}^{D} x_{ik} q_{kl} x_{jm} k_{ml})^{2p-2} \tag{A.47}$$

$$= (\sum_{k=1}^{D} \sum_{m=1}^{D} x_{ik} x_{jm} \sum_{l=1}^{d} q_{kl} k_{ml})^{2p-2} \tag{A.48}$$

$$= (\sum_{k=1}^{D} \sum_{m=1}^{D} x_{ik} x_{jm} a_{km})^{2p-2}, \tag{A.49}$$

where $a_{km} = \sum_{l=1}^{d} q_{kl} k_{ml}$. Let $z_{ij} = \sum_{k=1}^{D} \sum_{m=1}^{D} x_{ik} x_{jm} a_{km}$ Thus we have

$$\mathbb{E}(\|\frac{\partial (XQK^T X^T)^p}{\partial Q}\|_F^2) = p^2 \sum_{i=1}^{N} \sum_{j=1}^{N} \mathbb{E}(z_{ij}^{2p-2} \sum_{k=1}^{D} \sum_{l=1}^{d} (x_{ik} \sum_{m=1}^{D} x_{jm} k_{ml})^2) \tag{A.50}$$

$$= \sum_{i=1}^{N} \sum_{j=1}^{N} \sum_{k=1}^{D} \sum_{l=1}^{d} (\sum_{m=1}^{D} \mathbb{E}(z_{ij}^{2p-2} x_{ik}^2 x_{jm}^2 k_{ml}^2) \tag{A.51}$$

$$+ \sum_{m=1}^{D} \sum_{n=1,n\neq m}^{D} \mathbb{E}(z_{ij}^{2p-2} x_{ik}^2 x_{jm} k_{ml} x_{jn} k_{nl})) \tag{A.52}$$

$$= \sum_{i=1}^{N} \sum_{j=1}^{N} \sum_{k=1}^{D} \sum_{l=1}^{d} (\sum_{m=1}^{D} \mathbb{E}(z_{ij}^{2p-2} x_{ik}^2 x_{jm}^2 k_{ml}^2) \tag{A.53}$$

$$+ \sum_{m=1}^{D} \sum_{n=1,n\neq m}^{D} \mathbb{E}(z_{ij}^{2p-3} x_{ik}^2 x_{jm}^2 k_{ml}^2 x_{jn}^2 k_{nl}^2)) \tag{A.54}$$

$$\approx N^2 D d (D(D^{2p-2} d^{p-1} (2p-3)!! \sigma_x^{4p} \sigma_w^{4p-2}) + 0 \tag{A.55}$$

$$= N^2 D^{2p} d^p (2p-3)!! \sigma_x^{4p} \sigma_w^{4p-2} \tag{A.56}$$

showing that we can bound the gradient by a quantity of the form $\mathcal{O}(N)$ and the proof is complete.

$\square$

## A.2 EXPERIMENTS

### A.2.1 ABLATIONS

In this section we carry out ablations on the experiments we did in the main paper.

**Scale ablations:** The theory we developed in section 4.2 suggested that we needed to scale our polynomial activations by $\frac{1}{\sqrt{N}}$ to obtain a complexity bound of $\mathcal{O}(\sqrt{N})$. In general, we could also scale the polynomial activations by $\mathcal{O}(\frac{1}{\sqrt{N}})$ to see if we can get a better accuracr. Figure 10 carries out an ablation on both the ViT-Base architecture and the ViT-Small architecture to see how different scales affect the accuracy for the $x^3$ activation. We found that in general a scale from $\frac{1}{8}$ to $\frac{1}{25}$ seemed to perform very well.

**Activation ablations:** Our theory primarily compared the activations $\frac{1}{\sqrt{N}}x^3$ and $\frac{1}{\sqrt{N}}x$ with softmax where $N$ was the sequence length of the input. In this section we carry out further experimental comparisons of our activations with other activations. We compare with a variety of activations used in the literature as well as the exponential function $Exp = e^{-x}$ which can be thought of as softmax without the row normalization scaling. Table 5 shows the results of the experiments.

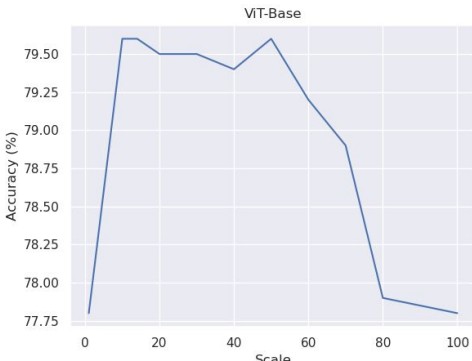 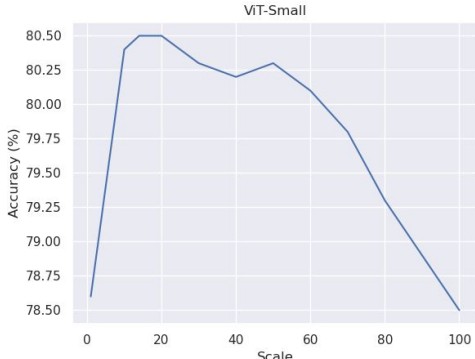

Figure 10: We show how the top-1% accuracy changes as the scale of the activation $x^3$ for a ViT-Base (left) and ViT-Small (right) architecture. The x-axis plots the denominator of the scale for easier readability. In other words as we go to the right of the x-axis the scale used on the activation $x^3$ is getting smaller.

| Activation | ViT-Base | ViT-small |
|:---:|:---:|:---:|
| softmax | 79.6 | 80.2 |
| $\frac{x^3}{14}$ | 79.6 | 80.5 |
| $\frac{x}{14}$ | 76.9 | 77.8 |
| $\frac{x^2}{14}$ | 78.7 | 79.9 |
| $ReLU$ | 77.3 | 77.5 |
| $ELU$ | 78.2 | 78.5 |
| $GELU$ | 78.2 | 78.4 |
| $Tanh$ | 77.1 | 77.3 |
| $Exp$ | 77.9 | 78.0 |

Table 5: Top-1% accuracy with various activations on a ViT-Base and ViT-Small architecture.

### A.2.2 FROBENIUS NORM COMPUTATIONS

In section 5.1 we showed plots of the Frobenius norm of the self-attention matrix and for the Jacobian of the self-attention matrix for different scalings of the $x^3$ and $x$ activations along with softmax. This was done for a ViT-Tiny architecture on the Tiny-ImageNet dataset. Figure 11 shows the plots of the Frobenius norm of the self-attention matrix for the Tiny-ViT architecture, during training, for all layers averaged over the heads within each layer. Figure 12 shows the Frobenius norm of the Jacobian of the self-attention matrix during training for each layer, averaged over the total number of heads within each layer.

Figures 13 and 14 show similar results for the ViT-Small architecture.

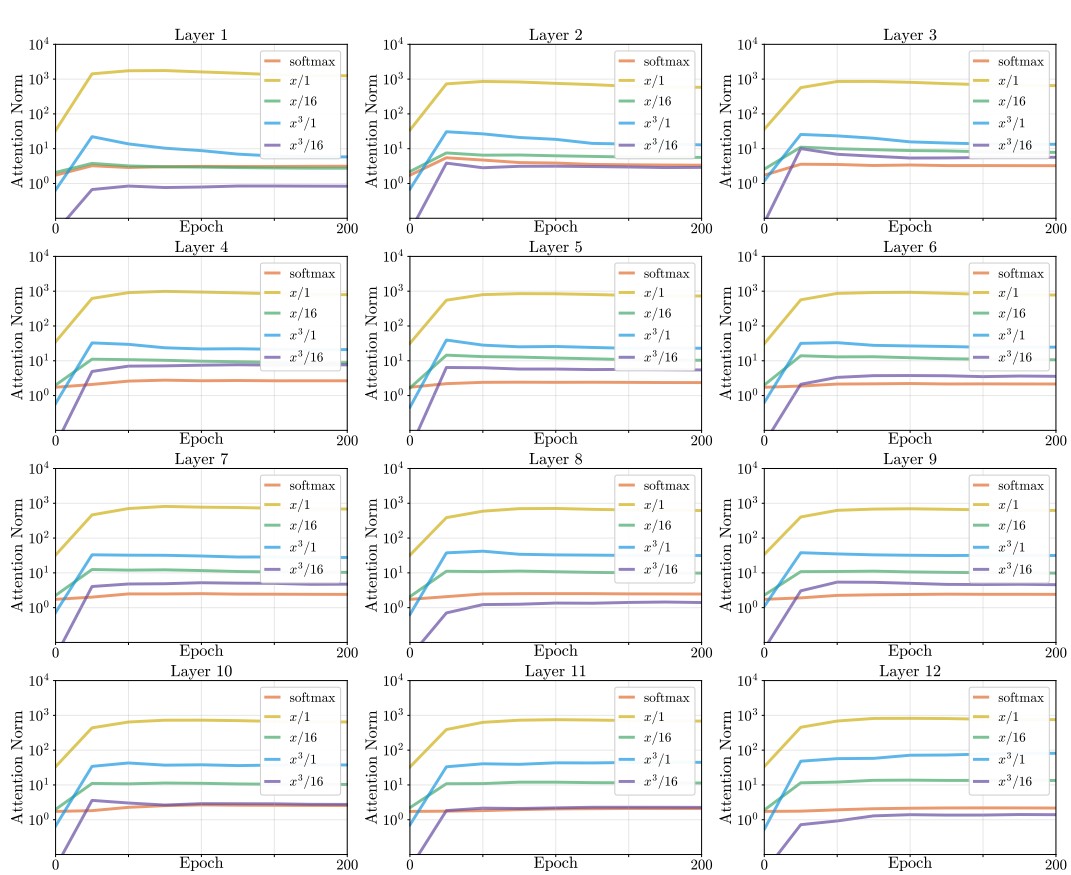

Figure 11: Frobenius norm of self-attention matrix for different scaled activations on ViT-Tiny during training (zoom in for better viewing).

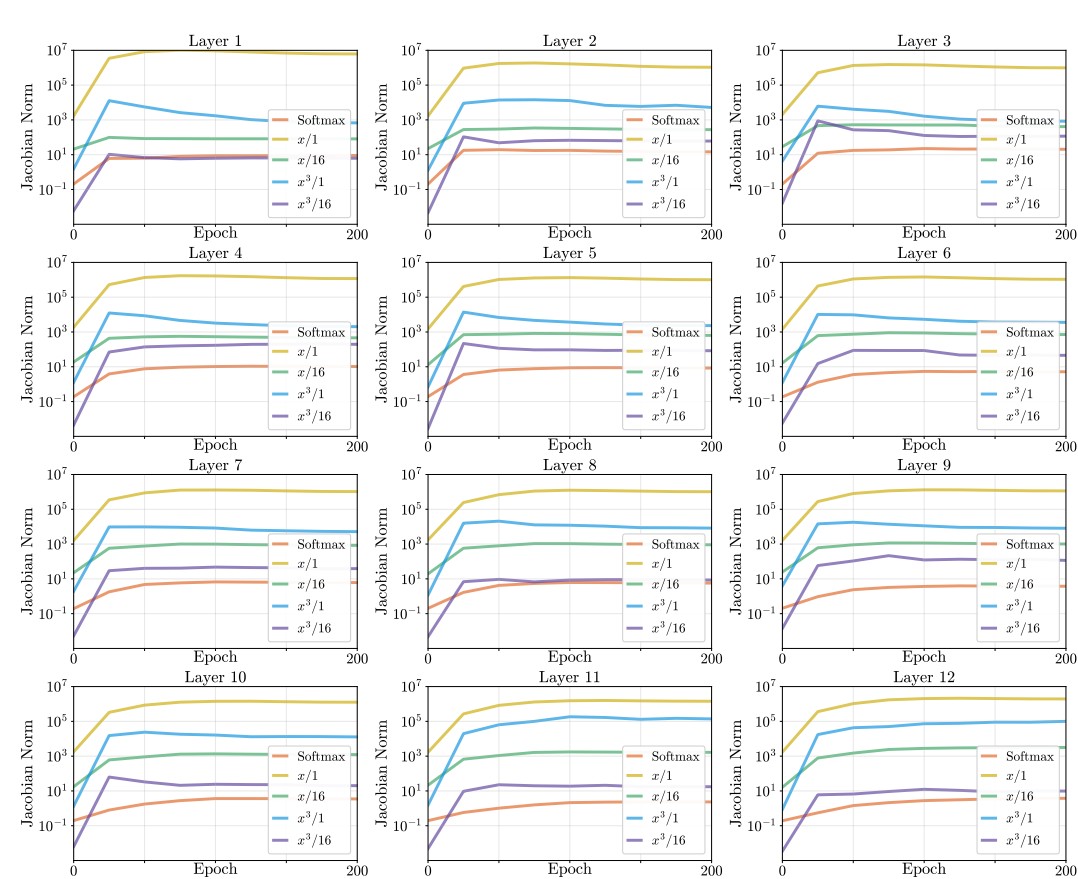

Figure 12: Frobenius norm of Jacobian of self-attention matrix for different scaled activations on ViT-Tiny during training (zoom in for better viewing).

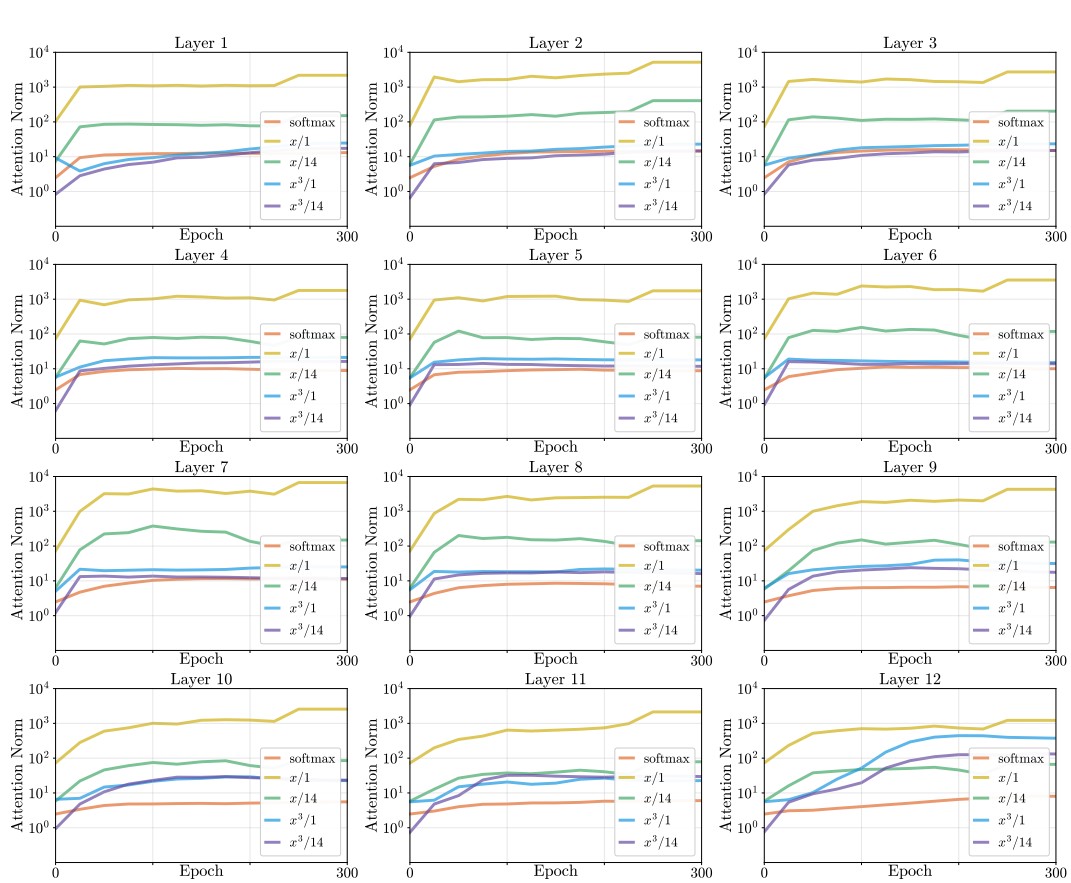

Figure 13: Frobenius norm of self-attention matrix for different scaled activations on ViT-Small during training (zoom in for better viewing).

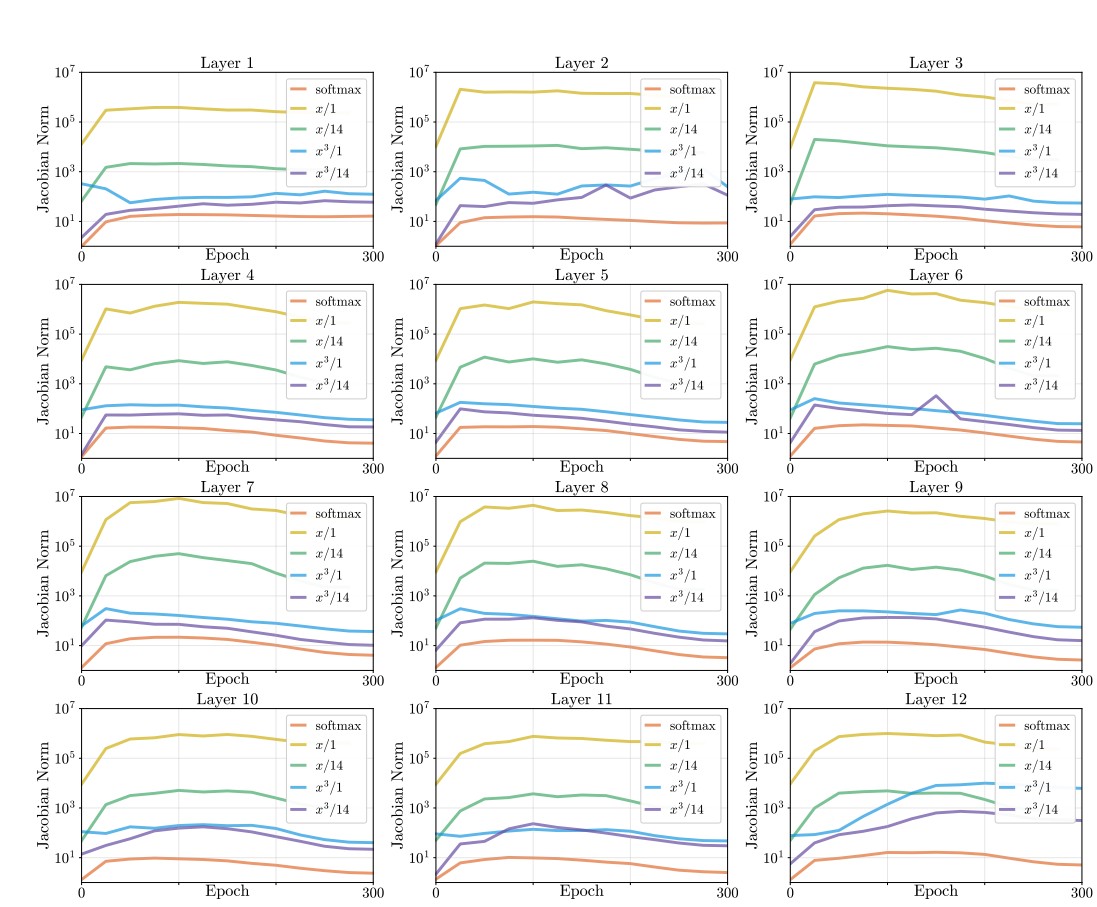

Figure 14: Frobenius norm of Jacobian of self-attention matrix for different scaled activations on ViT-Small during training (zoom in for better viewing).

