# OpenReview forum: "Rethinking Softmax: Self-Attention with Polynomial Activations"
_ICLR.cc/2025/Conference — ICLR 2025 Conference Withdrawn Submission_

### Official Review · Reviewer_kUcN · 2024-10-30

**Soundness:** 1
**Presentation:** 2
**Contribution:** 1
**Rating:** 3
**Confidence:** 4

**Summary:**

This paper attempts to develop a theory for why the row-normalised softmax (the standard form of softmax used in multi-head attention) really works in Transformer models. The authors claim that the usual reasons cited, namely providing a probability distribution (weights along a row sum to 1), non-negativity and sparsity do not explain the utility of the softmax function, but in fact row-normalised softmax regularises the (Frobenius) norm on the attention map and the linear gradient map of the attention map. The authors then develop some theory for how polynomial activation functions which break all three properties of softmax usually cited as reasons for success, but do regularise the norms of the attention map and its linear gradient map as per the theory derived.

The authors investigate the applicability of their theory by experimenting on image classification (ImageNet), object detection/instance segmentation (COCO) and NLP (Long Range Arena). Their experiments show that their polynomial activation functions in attention perform as well or slightly better than softmax. They provide some visualisations on the attention norm to back up their theory.

**Strengths:**

This paper addresses using polynomial functions which have been investigated before (e.g. Linear Transformers by Katharopoulos et al. investigate the use of 1st order polynomials, Babiloni et al. investigate 3rd order polynomials in Non-Local and Self-Attention layers), but the authors do try and break down some sacred cows regarding softmax which should be commended, they also look at the utility of softmax from a novel point of view (as far as I am aware). They experiment on vision tasks and language tasks which are good test beds for their method. The clearly show that not scaling polynomial activation functions is detrimental to performance.

**Weaknesses:**

Despite the interesting angle of their work I do have a number of criticisms.
1. The rigour with which the theories have been developed:
    1. Theorem 4.1 specifically Eq 4.2 which is proved in appendix 4.1.1, how does L719 work, I do not believe the factor of 2 be present, e.g. if $F_{i 1} = 1$ and all other values are 0 the equation gives leads to $1 - 0 = 2(1 + 0 + \ldots + 0) = 2$ which is obviously wrong.
    2. In Theorem 4.2, the authors have assumed X (hidden states), Q and K projections are i.i.d and therefore $A = XQ$ is has 0 covariance with $B^{T} = KX$, I would argue this is an extremely strong assumption except in the trivial cases and so is not applicable to Transformers. I therefore question the utility of the theory developed on this basis. In the general case $\text{Cov}(A, B) = Q K^\top$ when X is drawn from $\mathcal{N}(0, 1)$.
    3. In Theorem 4.2, the authors comes to final terms which contain both $N$ (the sequence length) and $D$ (the feature dimension) e.g. Eq A.41 and then claim the quantity grows with $\mathcal{O}\left(\sqrt{N}\right)$. This will only be the case when $N \gg D$. Similar arguments can be made for all of Theorem 4.2, Corollary 4.3, Theorem 4.4 and Corollary 4.5. Noting that for ViT-Base $D = 768$ and $N = 14^2 = 196$ (https://huggingface.co/google/vit-base-patch16-224/blob/main/config.json). Therefore the much larger condition cannot be satisfied. As a side note A.1.2 introduces a parameter $d$ (not $D$) which is not defined, perhaps this is the dimension of a single attention head but it is not clear.

2. The linkage between the derived theory and the experiments.
    1. L201-207 say that corollaries 4.3 and 4.5 show that scaled polynomial activation functions can perform similarly to softmax. It is not made clear why this is the case unless simply because the relationship between the activation function use and norm both grow with $\sqrt{N}$.
    2. Figure 1 and L240-245. If you argue that polynomial activation functions should be scaled by $\frac{1}{\sqrt{N}}$, but then say as the sequence length *decreases* the scale applied should *decrease*. Should this not be the opposite i.e. the scale applied should increase?
$N = 256, 64, 16, 8 \to \frac{1}{\sqrt{N}} = 0.125, 0.0625, 0.25, 0.353. Perhaps I have misunderstood the argument, but regardless it is not clear.

3. Significance of the attention map visualisations?
    1. Figure 6 and Figure 7 show pairs of attention map visualisations for specific heads for softmax and polynomial activations. The arguments made here about "contrast" and "difference" seem obvious given the function is so different. E.g. softmax maps cannot be negative. Moreover, there is no pre-ordained behaviour of a specific attention head and so I do not understand what point the authors are trying to make other than the activation functions are different.

4. Natural language processing sequence length.
Transformers for NLP make use of causal masking and so the effective sequence length for each query is different (incremented by 1) and this has not been considered in the experiment, to match the theory shouldn't the activation scaling evolve as the effective sequence length increases?

**Questions:**

The questions I have naturally arise from the weaknesses I have listed above. However concretely:

Here’s a breakdown of your critique into concise, specific questions:

1. **Rigour of Theoretical Development**
   - How does L719 work in Theorem 4.1, specifically Equation 4.2? Should the factor of 2 be present?
   - In Theorem 4.2, why do the authors assume $X$, $Q$, and $K$ are i.i.d., making $A = XQ$ uncorrelated with $B^T = KX$? Is this assumption valid for Transformers?
   - Should the covariance of \(A\) and $B$ in Theorem 4.2 be \(Q K^\top\) when \(X\) is drawn from \(\mathcal{N}(0, 1)\) instead?
   - In Theorem 4.2, can the final terms containing both $N$ (sequence length) and $D$ (feature dimension) grow as $\mathcal{O}(\sqrt{N})\$ if $N$ is not much larger than $D$?
   - Is the parameter $d$ in A.1.2 meant to represent the dimension of a single attention head? If not, please define it.

2. **Linkage Between Theory and Experiments**
   - How do Corollaries 4.3 and 4.5 demonstrate that scaled polynomial activation functions perform similarly to softmax?
   - If polynomial activation functions should be scaled by $\frac{1}{\sqrt{N}}$, why does the paper suggest the scale should decrease as sequence length decreases? Shouldn’t the scale increase?

3. **Significance of Attention Map Visualisations**
   - What is the purpose of showing attention map visualisations in Figures 6 and 7 if the contrast between softmax and polynomial activations is already expected due to the fundamental differences between the functions?

4. **NLP Sequence Length and Scaling**
   - Should the activation scaling change to account for the varying effective sequence length caused by causal masking in NLP applications?

---

### Official Review · Reviewer_Dajh · 2024-11-01

**Soundness:** 3
**Presentation:** 4
**Contribution:** 2
**Rating:** 6
**Confidence:** 3

**Summary:**

Analyzes the softmax activation in Transformer attention, and shows that using a different activation with a 1/sqrt(N) scale matches Softmax. This is shown from a theoretic perspective, and then confirmed using experiments in both vision (classification and object detection) and NLP domains.

**Strengths:**

1. Good writing and presentation. Both the proofs and experiments are well presented, easy to read, while concise.
2. Shares all theoretic proofs with detailed explanation of intermediate steps, then confirms the theoretic results with experiments. Experimental results align with the theory.
3. Uses multiple vision and NLP experiments to show the theory holds across domains.

**Weaknesses:**

- Does not mention what to do with dynamic sequence length tasks (e.g. language generation). Do you dynamically scale during both training and inference dependent on current sequence length (or do you take e.g. maximum during training)?

- Seems to mainly focus on ViT, lacking LM experiments (even at small scale). As seen, ViT is a lot more flexible and even gets decent results without doing the attention scaling. The NLP experiments are very small, and lack details. Experiments are minimal.

- Lacking details for the NLP experiments. No mention of model size, hyperparameters overview/recipe and information about the tasks themselves. Does refer to a work ([https://ojs.aaai.org/index.php/AAAI/article/view/17664](https://ojs.aaai.org/index.php/AAAI/article/view/17664)), but does not mention what they use from that work.

- Does not cite [https://arxiv.org/abs/2409.04431](https://arxiv.org/abs/2409.04431), which this works seems to have a lot of overlap with. The focus of this work is different, but especially section 3.2 (Regularity of Sigmoid Attention) has a lot of overlap.

- Typos

    - “accuracr” (A.2.1) -> accuracy

**Questions:**

- Can you add details on what to do with dynamic sequence length (while training/inference)?

- Can you add small LM experiments?

- Can you add details on why the training with activations x^3 and x did not work in experiment section 5.2? Was the training unstable? Did the loss explode, did it stay flat?

---

### Official Review · Reviewer_SENZ · 2024-11-04

**Soundness:** 1
**Presentation:** 2
**Contribution:** 1
**Rating:** 3
**Confidence:** 3

**Summary:**

This paper shows that softmax activation implicitly regularizes the Frobenius norm of the attention matrix in transformer models during training. It proposes using simple scaled polynomial activations as alternatives for similar regularization effects. Experiments across vision and NLP tasks show that these polynomial activations can match or outperform softmax attention.

**Strengths:**

The paper is easy to read and experiments are across multiple domains and on various tasks. The paper gives an theoretical analysis of scaled-polynomial activation and attention norm and attention's jacobian matrix norm

**Weaknesses:**

1. Motivation:
Softmax stabilizes the gradient but also causes gradient vanishing (the gradient is less than 1 and saturates to 0, see [1, 2]). Therefore, i disagree with the argument in the paper contribution stating that: "we theoretically show that softmax has a regularization effect on attention and argue this plays
a crucial role in its success."

In addition, in their theory, there is only the analysis of softmax and polynomial activation (used in attention) but I have not found the theoretical analysis that shows that this regularized effect correlates with the performance of transformer models.

In the experimental results, I also fail to see the same correlation. For example, in Figures 2 and 3, the attention norm and Jacobian norm of models using (x^3/1) are constantly lower than that of models using (x/16). However, in Table 1, the model with (x^3/1) has significantly worse performance than the model with (1/16).

The paper also states: "We advise the reader that this paper diverges from the usual pursuit of creating cutting-edge transformer architectures for achieving state-of-the-art results on benchmark datasets. Instead, our focus is on critically examining softmax attention to determine whether its effectiveness is a result of true interpretability or a more nuanced regularization mechanism". As far as I understand, the paper aims to examine the reason why behind the use of softmax activation in attention, less about aiming to improve it. However, I think the paper fails to argue why regularization is the source of softmax attention success and their proposed polynomial activation also fails to significantly improve over softmax.

2. Not convincing experiment results
Again, as I pointed out above, the experiments do not convince me how the norm-regularized effect correlates with the performance of transformer models. In addition, in Table 2, ViT-base and ViT-small  (x/14) have lower performance than (x^3) and that seems to correlate with norm analysis in Figures 13 and 14 (where Vit-models x/14 have higher attention and Jacobian norm than those of ViT-models x^3). However, also in Table 2, Deit-base and -small (x/14) outperform the rest. However, no norm analysis of this model is shown in the experiment.

Most of the improvements in performance made by polynomial activation are only marginal

LRA benchmark is not an NLP benchmark, 3 out of 5 tasks are not NLP, including Listops, Image Classification and pathfinder. The authors should not make this mistake, and they should understand the benchmark that they are testing on.

3. Small mistakes in the proof: from the line 665-669, the author makes a replicate typo in $e^{2a_{11}}$ and $e^{2a_{NN}}$.

*Reference*
[1]. Oliver Paul Richter and Roger Wattenhofer. Normalized attention without probability cage, 2022.
[2]. Escaping the gradient vanishing: Periodic alternatives of
softmax in attention mechanism

**Questions:**

1. Can you show more why the norm-regularization is very important for softmax attention?
2. Can you show the same of figure 2, 3 for Deit-small and deit-base?

---

### Official Review · Reviewer_pynZ · 2024-11-06

**Soundness:** 3
**Presentation:** 3
**Contribution:** 3
**Rating:** 6
**Confidence:** 4

**Summary:**

The paper presents an in-depth investigation into the role of softmax in self-attention, demonstrating that its success is due to the implicit regularization of the attention matrix norm. Additionally, the authors propose a polynomial activation function as an alternative to softmax, aiming to achieve a similar effect within the self-attention mechanism.

**Strengths:**

1. The paper conducts in-depth exploration of the role of softmax in the self-attention
2. The polynomial activation proposed by paper is novel and justified theorectically.
3. The experimental results show promising performance.

**Weaknesses:**

1. The proposed polynomial activation method does not seem to achieve the promising results observed in vision tasks. Do the authors have any insights into why this might be the case?
2. The paper only adopts $x$ and $x^3$ as the polynomial activation functions. Is there a particular reason why other values were not considered?
3.  The paper lacks a discussion on the computational overhead. Given that the proposed method introduces a new form of self-attention, it would be beneficial to include an analysis of time complexity or provide a comparison of actual running times between the softmax-based and polynomial-based methods.
4. On line 255, $\frac{1}{8}x^3$ should be $\frac{1}{16}x^3$. The same mistake in the caption of Table 1.

**Questions:**

See weakness

---

### Note · Authors · 2024-12-02

I have read and agree with the venue's withdrawal policy on behalf of myself and my co-authors.